# SMUG: Sand Mixing for Unobserved Class Detection in Graph Few-Shot Learning

## ABSTRACT

Graph few-shot learning (GFSL) has achieved great success in node classification tasks with rare labels. However, graph few-shot classification (GFSC) models often encounter the problem of classifying test samples with unobserved (or unknown) classes due to the rareness of labels. We formulate this problem as out-of-distribution (OOD) sample detection in inductive graph few-shot learning. This paper presents SMUG, a novel GFSL framework that can detect unobserved classes. Since we have no ground-truth OOD samples in a practical training dataset, it is challenging for the GFSC model to retrieve knowledge about unknown classes from labeled samples. To address this difficulty, we propose a sand mixing scheme to introduce observed classes as artificial OOD samples into meta-tasks. We also develop two unsupervised OOD discriminators to identify OOD samples. Thus, we can assess the performance of OOD discriminators since we know the true classes of these artificial OOD samples. Subsequently, we design a novel training procedure to optimize the encoder based on the performance of the OOD discriminators and the GFSC model. It not only enables the GFSL model to distinguish OOD samples but also promotes the classification accuracy of normal samples. We conduct extensive experiments to evaluate the effectiveness of SMUG based on four benchmark datasets. Experimental results demonstrate that SMUG achieves superior performance over state-of-the-art approaches in OOD detection and node classification.

## KEYWORDS

graph few-shot learning, out-of-distribution detection, sand mixing

**ACM Reference Format:**

Anonymous Author(s). 2018. SMUG: Sand Mixing for Unobserved Class Detection in Graph Few-Shot Learning. In *Proceedings of Make sure to enter the correct conference title from your rights confirmation emai (Conference acronym 'XX).* ACM, New York, NY, USA, 10 pages. https://doi.org/XXXXXXX.XXXXXXX

## 1 INTRODUCTION

Graph node classification has attracted much attention as an essential task in Web data mining. In practice, many classes have limited labeled instances, rendering a long-tailed distribution of class samples. Nevertheless, deep learning models require many

labeled samples for model training. Therefore, learning with a handful of labeled samples is a key challenge in translating the research efforts of deep learning to real-world applications. To tackle this difficulty, graph few-shot learning (GFSL) has been proposed to enable a model to learn new tasks from a few labeled samples [1, 2].

Most graph few-shot classification (GFSC) models exploit the concept of meta-learning [3–5], which comprises a series of meta-tasks constituted by meta-training and meta-test. Each meta-task has a support set and a query set. A GFSC model is trained through meta-training tasks, which know the labels of samples in the support and query sets. A meta-test task only knows the labels of samples in the support set. Given a meta-test task, a trained GFSC model classifies the samples in the query set by transferring the learned class distributions in the support set [1].

It is generally assumed that samples in the support and query sets have common classes for a single meta-task [6–8]. However, some samples in the query set of a meta-test task may actually not belong to any known classes in the support set in practical applications [9, 10]. For example, some rare cases never observed in medicine could be collected in a query set, or papers about a new research topic could gradually emerge in a citation graph. We define an out-of-distribution (OOD) sample as one in the query set that does not belong to any known classes in the support set for a meta-test task. A conventional GFSC model could mistakenly classify an OOD sample into an existing category, sometimes resulting in severe incidents in safety-critical applications. Therefore, it is desirable to design a GFSC model that can detect OOD samples.

GFSC models without the awareness of OOD samples will categorize an unobserved class sample into existing classes. These models face two dilemmas: i) OOD samples are mistakenly assigned to an arbitrary category, and ii) the existence of OOD samples harms the classification of normal samples. Meanwhile, models with the awareness of OOD samples can better categorize normal samples into their proper classes if we can identify OOD samples. We define the problem of identifying OOD samples in GFSC tasks as Few-shot Classification with Out-of-Distribution Detection (GFSC-OOD). Conventional OOD detection methods have achieved promising results [11–13]. However, applying these methods directly to the GFSC-OOD problem is difficult due to the lack of labeled data. Moreover, it poses more difficulty in learning the distributions of normal samples in the existence of OOD samples.

Most previous studies employ an independent OOD discriminator to distinguish OOD samples from normal ones by modeling it as a binary classification problem [9, 14, 15]. Although training an OOD discriminator independently distinguishes OOD samples from normal ones, it pulls different classes of normal samples closer, degrading the classification accuracy. We observe that GFSC and OOD detection can benefit from each other. A good GFSC model should encode an OOD sample far away from normal ones in the

latent space. On the other hand, the performance of the GFSC model can be promoted if OOD samples are correctly identified.

This paper presents SMUG, a novel GFSC framework to address the GFSC-OOD problem. SMUG equips existing GFSC models with the ability of OOD detection and promotes classification performance. However, we usually have no ground-truth OOD samples in a practice dataset. To solve the cold-start problem, we propose a sand mixing scheme to introduce some known classes as artificial OOD samples. We mix these artificial OOD samples into the query sets of meta-training tasks. It allows us to assess the performance of the OOD discriminators in the training process because the true labels of artificial OOD samples are known.

We design two unsupervised OOD discriminators to identify OOD samples with distance-based and probability-based criteria, respectively. Many existing distance-based methods judge OOD samples based on the minimum distance between a sample and class prototypes, ignoring the sample distributions of different categories. We propose a novel distance-based discriminator to determine a test (or query) sample as OOD based on the support radii of different classes, which are closely relevant to the distributions of specific classes. Most existing probability-based approaches set an explicit threshold for OOD detection. Differently, we assume that the probability of an OOD sample being classified into any normal class is small and even. Accordingly, we propose an adaptive thresholding method for the probability-based discriminator.

We train an inductive graph neural network (GNN)-based encoder to map nodes into a low dimensional space. The optimization of the encoder affects the performance of the GFSC model and the OOD discriminator. Training the encoder solely based on the performance of the GFSC model will reduce the performance of OOD detection. Nevertheless, training the encoder solely based on the performance of the discriminator will lead to the collapse of normal nodes, i.e., the encoder pulls different classes of normal samples closely, significantly degrading the classification performance of normal samples. To address this issue, we design a novel training procedure to optimize the encoder by integrating the performance of the OOD discriminator as a weak signal with the performance of the FSC model. This scheme not only enables the optimized model to identify OOD samples but also enhances the performance of the classification task by learning better distributions of normal samples. The contributions of this work are as follows:

- We propose a novel GFSL framework to equip existing GFSC models with the ability of OOD detection. Unlike conventional GFSC models, our method incorporates an unsupervised OOD discriminator to detect OOD samples.
- We propose a sand mixing scheme to solve the cold-start training problem caused by the absence of ground-truth OOD samples. It allows us to assess the performance of OOD discriminators, which are employed as a weak signal to train the encoder.
- We develop two OOD detection discriminators and design two corresponding loss functions to optimize the encoder based on the performance of OOD discriminators.
- We conduct experiments to evaluate the effectiveness of the proposed method. Experimental results show that our proposed method outperforms runner-up methods by a margin of 2.2% and 1.6% in ACC and F1-score.

The rest of this work is organized as follows. Section 2 presents the related work in graph few-shot learning and out-of-distribution detection. Section 3 introduces the preliminaries of few-shot classification and the definition of the GFSC-OOD problem. Section 4 shows the details of our method. Section 5 demonstrates the experiments conducted to evaluate the effectiveness of the proposed method. Finally, Section 6 concludes the work and discusses possible research directions in future work.

## 2 RELATED WORK

### 2.1 Graph Few-shot Learning

Few-shot learning (FSL) has received much attention due to its effectiveness in encoding data with rare labels [1, 2, 16–18]. Generally, existing FSL models fall into three categories: fine-tune-based, data augmentation, and transfer learning-based methods. Fine-tune-based approaches pre-train a model on a large source dataset and then fine-tune the model based on the target dataset [19, 20]. These methods suit situations where the target and source datasets have similar distributions. When the target and source datasets have different distributions, the trained model usually over-fits the target dataset because a small amount of data cannot reflect the real distribution well.

Data augmentation-based methods consider the fundamental problem of FSL as the low diversity of samples due to small amounts of samples [21]. These approaches employ data expansion [22] or feature augmentation [23] to improve sample diversity with auxiliary information. Data expansion adds new data that contains unlabeled data [24, 25] or synthetic labeled data [26, 27] to the original dataset. Feature augmentation improves sample diversity by enhancing the features with good generalization [28, 29].

Transfer learning-based methods refer to using old knowledge to assist in learning new knowledge [30]. Data in the source and target domains should have certain correlations. In general, the stronger the correlation, the better the effect of transfer learning [31]. These approaches can be divided into two groups: metric-learning-based and meta-learning-based. Metric learning [32], also called similarity learning, classifies samples by calculating the distance between a sample to be classified and the known classes [33–35]. Meta-learning is also called learning to learn [36], which learns meta-knowledge from many prior tasks so that the model can learn faster in new tasks. Currently, meta-learning methods have transformed from one-shot learning [27, 33] to few-shot learning [3, 4, 7].

Graph neural networks (GNNs) are essential for learning node representations of graph data. Common GNNs include graph convolutional networks [37], gated graph neural networks [38], and graph attention networks [39]. Most existing graph few-shot learning (GFSL) models employ GNNs as an encoder to learn node representations and exploit the meta-learning procedure for model training. However, few models consider the out-of-distribution (OOD) detection problem, which is common in GFSL due to the lack of labeled data and the dynamic nature of graph data.

### 2.2 Out-Of-Distribution Detection

OOD detection refers to detecting samples that do not belong to any known classes in the training set. Traditional machine learning methods assume that samples in the training and test datasets

are independent and identically distributed (i.i.d.), *i.e.*, they are in-distribution (ID) samples. In practice, the test data received by the model may contain some OOD samples. Existing models often assign OOD samples to some known classes, leading to the limited applications of these models to safety-critical fields such as medicine, finance, and autopilot. Thus, enabling a model to identify OOD samples is essential to the safety of machine learning models.

There are three kinds of OOD detection methods: classification-based methods, probability-based approaches, and distance-based schemes. Classification-based methods use the maximum softmax probability as an indicator score of OOD-ness [40]. Most models focus on deriving an OOD score based on deep neural networks [41–43]. Probability-based approaches characterize ID samples with probabilistic models and identify samples in low-density regions as OOD. A class-conditional Gaussian distribution explicitly models ID samples and identifies OOD samples based on their likelihoods [44]. However, some works find that probabilistic models sometimes assign higher likelihoods for OOD samples [45, 46]. Several researchers attempt to solve the problem using likelihood regret [47] and density models [45]. However, these probabilistic models are prohibitively hard to train and optimize. Distance-based methods assume that OOD samples are relatively far from the prototypes of ID classes [48]. Therefore, they identify OOD samples based on the distances to the prototypes of ID classes [49].

With the rapid development of FSL models, some OOD detection methods have been studied for FSL [11, 15, 50]. Most of these methods aim to design an OOD detection model independent of the FSC model, ignoring the promotion of OOD detection for FSC models. Unlike these approaches, we employ the performance of OOD discriminators as a weak signal to supervise the training of the FSL model.

## 3 PRELIMINARIES

### 3.1 Graph Few-shot Classification

Denote a graph as $\mathcal{G} = \{\mathcal{X}, \mathcal{E}, \mathbf{X}'\}$, where $\mathcal{X}$ is the set of nodes, $\mathcal{E}$ is the set of edges, and $\mathbf{X}'$ is the raw feature matrix of nodes. The node set $\mathcal{X} = \mathcal{X}^L \cup \mathcal{X}^U$ constitutes two kinds of nodes: a labeled set $\mathcal{X}^L$ with ground-truth labels $\mathbf{y}^L = [y_1, y_2, ...y_{|\mathcal{X}^L|}]$ and an unlabeled set $\mathcal{X}^U$. The classification problem is to learn a classifier $g_{\Theta'}: \mathcal{X} \rightarrow \mathbf{y}$ based on the patterns learned from $\mathcal{X}^L$, and $\Theta'$ represents the model parameters. The classifier then categorizes the nodes in $\mathcal{X}^U$.

A general solution is to learn an encoder $f_\Theta: \mathcal{X} \rightarrow \mathbf{X}^{n \times d}$, where $n$ is the number of samples, $\Theta$ represents the model parameters, and $d$ represents the embedding dimensionality. The encoder $f_\Theta$ transforms nodes into a latent vector space, preserving the structural and attributive information of $\mathcal{G}$. The learned representations $\mathbf{X}$ are then used to train a classifier $g_{\Theta'}$ such as support vector machine (SVM), logistic regression, and multi-layer perceptron (MLP). This approach requires a mass of labeled samples for model training and fails to handle scenarios with rare labels.

Graph few-shot learning (GFSL) addresses the above problem by decomposing the classification task into multiple meta-tasks $\mathcal{T} = \mathcal{T}^{train} \cup \mathcal{T}^{test}$, where $\mathcal{T}^{train} = \{\mathcal{T}_1, \mathcal{T}_2, ..., \mathcal{T}_T\}^{train}$ is the set of meta-training tasks, $T$ is the number of tasks, and $\mathcal{T}^{test}$ is the set of meta-test tasks. Each meta-task $\mathcal{T}_t = (\mathcal{S}_t, \mathcal{Q}_t)$ contains two sampled mini-sets, including a support set $\mathcal{S}_t$ and a query

set $\mathcal{Q}_t$. A support set $\mathcal{S}_t = \{(x_1, y_1), (x_2, y_2), ..., (x_{N \times K}, y_{N \times K})\}$ is generated by randomly selecting $N$ classes from the class set $C$ and uniformly picking $K$ samples for each category of the $N$ classes from $\mathcal{X}^L$. A query set $\mathcal{Q}_t = \{(x_1, y_1), (x_2, y_2), ..., (x_{N \times M}, y_{N \times M})\}$ also contains the same $N$ classes, and each class is composed of $M$ samples extracted from the remaining samples. The model learns an encoder $f_\Theta$ based on the meta-training tasks. It minimizes the loss of the predictions for samples in the query sets by contrasting them with those in the support sets. Given a meta-task $\mathcal{T}_t$, a common loss is usually formulated as follows:

$$L_t = \sum_{k=1}^{N \times M} L_t^k, \tag{1}$$

where $L_t^k = -\sum_{i=1}^{N} \log \frac{\exp(-dist(f_\Theta(x_k), \bar{\mathbf{x}}^i))}{\sum_{j=1}^{N} \exp(-dist(f_\Theta(x_k), \bar{\mathbf{x}}^j))}$ is the loss of the $k$-th sample, $\bar{\mathbf{x}}^i = \sum_{(x, y=i) \in \mathcal{S}_t} f_\Theta(x)/K$ and $\bar{\mathbf{x}}^j$ are the prototypes of the $i$-th and $j$-th classes in the support set, respectively; and $dist(\mathbf{x}_k, \bar{\mathbf{x}}^i)$ is the distance from $\mathbf{x}_k$ to $\bar{\mathbf{x}}^i$.

In this way, the model learns meta-knowledge gradually from the training set and transfers them to the meta-test tasks $\mathcal{T}^{test} = (\mathcal{S}^{test}, \mathcal{Q}^{test})$, where the support set $\mathcal{S}^{test}$ contains a small number of labeled samples. Besides, samples in $\mathcal{S}^{test}$ and $\mathcal{Q}^{test}$ are unnecessary to belong to any classes in $C$. The obtained model is then used to predict the labels of samples in the query set $\mathcal{Q}^{test}$ by contrasting them with those in the support set $\mathcal{S}^{test}$. This problem is named the $N$-way $K$-shot classification problem.

### 3.2 Problem Definition

It is generally assumed that samples in the support set $\mathcal{S}_t$ and the query set $\mathcal{Q}_t$ share common classes so that samples in $\mathcal{Q}_t$ can be correctly predicted by learning the distributions of the samples in $\mathcal{S}_t$. Besides, samples in the training and test sets are supposed to be independent and identically distributed (*i.i.d.*), and all samples in $\mathcal{X}$ obey a common distribution $x \sim P$. In practice, the sample set $\mathcal{X}$ may contain some nodes that do not belong to any known classes, *e.g.*, a few recently published papers about an emerging topic, and some rare disease cases that have never been observed in medicine. We call these samples as out-of-distribution (OOD) samples and redefine $\mathcal{X}$ as $\mathcal{X}' = \mathcal{ID} \cup \mathcal{OOD}$. In GFSC models, the query set of a meta-test task is likely to contain some OOD samples.

This paper aims to train a graph encoder to distinguish OOD samples from normal ones in a meta-task and improve the classification performance of a GFSC model. Particularly, we would like to optimize the encoder based on the performance of both the OOD discriminator and the GFSC model, i.e., the encoder maps OOD samples far away from normal ones while improving the distinction of normal samples.

## 4 THE PROPOSED METHOD

This section introduces SMUG, which powers graph few-shot learning models with the ability to identify samples with unobserved classes and improves the performance of GFSC models.

### 4.1 An Overview

Fig. 1 presents an overview of SMUG, which comprises two training phases. The first phase involves a common training process of a GFSC model. A series of meta-tasks are generated by sampling from

the training set. For each meta-task, nodes in a support and query set pair are fed into an encoder $f_\Theta$ to learn their low-dimensional representations. This paper employs an inductive graph convolutional network (GCN) encoder [51] for graph representation learning. Then, the obtained embeddings are fed into a multi-layer perceptron (MLP) as the classifier $g_{\Theta'}$. The cross-entropy loss is computed according to the classification results of query nodes, and gradient descent is employed to update the parameters.

The second phase is our key contribution. The main idea is to mix the query sets of meta-training tasks with artificial OOD samples (sand mixing), which are then identified before being fed into the MLP classifier. Specifically, we randomly select some known classes in the training set as OOD samples. Then, a proportion $\varphi$ of meta-training tasks are randomly selected, and sand mixing is performed by adding artificial OOD samples to the query sets of these meta-tasks. The mixed and clean meta-tasks are fed into the encoder for representation learning. Unlike conventional FSC models, an unsupervised OOD discriminator is employed to identify OOD samples. The challenge lies in finding an effective criterion to distinguish OOD samples from normal ones. In this paper, we propose two discriminators to identify OOD samples. The identified OOD samples are then filtered out and no longer involved in subsequent classification tasks. We also design two corresponding OOD losses for the two discriminators and feed them back to the encoder. The graph encoder is optimized based on a combination loss of the OOD loss and the classification loss. It enables the encoder to learn the distributions of both normal and OOD samples.

## 4.2 Graph Convolutional Network Encoders

In order to handle dynamic graphs, we employ an inductive graph convolutional network (GCN) as the encoder [51]. The GCN encoder follows a neighborhood aggregation scheme, where a node aggregates information from its local neighborhood at each layer to update its representation. A GCN layer is formulated as follows:

$$\mathbf{h}_v^l = \sigma\left(\mathbf{W}^l \cdot \left(\mathbf{h}_v^{l-1} \oplus \text{AGGR}\left(\left\{\mathbf{h}_u^{l-1}, \forall u \in \mathcal{N}_{(v)}\right\}\right)\right)\right), \quad (2)$$

where $l \in \{1, 2, \ldots, L\}$ is the layer indicator, $L$ is the number of layers, $\sigma$ is an activation function, $\oplus$ represents the concatenation function that combines the aggregated neighborhood vectors at the $l$-th layer with the node vector of $v$ at the $(l-1)$-th layer, $\mathbf{h}_v^l$ denotes the node representation of node $v$ at the $l$-th layer, $\mathbf{h}_v^0$ is the raw feature of node $v$, $\mathcal{N}_{(v)}$ denotes the set of $v$'s neighbors, AGGR is a function aggregating the neighbor vectors, and a set of trainable weight matrices $W^l$ is the core for message propagation.

Node representations $\mathbf{X}$ are obtained by stacking the outputs of every layer in the GCN encoder:

$$\mathbf{X} = [\mathbf{H}^0, \mathbf{H}^1, \ldots, \mathbf{H}^L], \quad (3)$$

where $\mathbf{H}^l$ is the outputs of the $l$-th layer for all nodes. We use $f_\Theta(\cdot)$ to denote the GCN encoder, where $\Theta = \left\{\mathbf{W}^l, l \in \{0, 1, \ldots, L\}\right\}$ are the set of trainable parameters.

## 4.3 Meta-task Settings

Generally, a GFSC model learns over diverse meta-training tasks and transfers learned knowledge to the target meta-test tasks. In each training episode, an $N$-way $K$-shot meta-task $\mathcal{T}$ is constructed as follows:

$$\begin{cases} \mathcal{T} = (\mathcal{S}, \mathcal{Q}) \\ \mathcal{S} = \{s_1, s_2, \ldots, s_{N \times K}\} = \bigcup_{c=1}^N \mathcal{S}_c \\ \mathcal{Q} = \{q_1, q_2, \ldots q_{N \times M}\} \end{cases}, \quad (4)$$

where $s$ and $q$ are samples in the support and query sets, respectively; $\mathcal{S}_c = \{s_1^c, s_2^c, \ldots, s_K^c\}$ is the set of samples belonging to class $c$, and $s_i^c$ represents the $i$-th sample of the $c$-th class in the support set. We assume that $\mathcal{S}_c$ contains no OOD samples because samples in the support set are usually labeled manually according to strict rules. To address the cold-start problem, we randomly select some labeled classes as artificial OOD samples. We add these OOD samples to the query sets of meta-training tasks with a probability of $\varphi$ to simulate OOD samples in real applications. We name such a process sand mixing. The new meta-task after sand mixing is:

$$\begin{cases} \mathcal{T}' = (\mathcal{S}, \mathcal{Q}') \\ \mathcal{Q}' = \mathcal{Q} \cup \mathcal{Q}^{OOD} \end{cases}, \quad (5)$$

where $\mathcal{Q}^{OOD} = \{q_1^{OOD}, q_2^{OOD}, \ldots q_\Omega^{OOD}\}$, and $\Omega$ is an adjustable parameter representing the number of OOD samples. It is worth noting that we know the labels of these OOD samples.

## 4.4 OOD Discriminator

The encoder $f_\Theta$ maps both nodes in $\mathcal{S}$ and $\mathcal{Q}'$ into a latent vector space. For an $N$-way $K$-shot support set $\mathcal{S}$, the encoder $f_\Theta$ is expected to map intra-class samples with close distances while pushing inter-class samples far away. A key challenge is to find a criterion for distinguishing OOD samples from normal ones because the model knows nothing about the distributions of OOD samples except for their existence.

We propose two discriminators to determine whether a test sample $q$ is an OOD sample, including a distance-based and a probability-based criterion. Fig. 2 illustrates the criteria of the two discriminators. We use a prototype vector to represent the cluster embedding of samples in $\mathcal{S}_c$ for each category. For the distance-based discriminator, a test (or query) sample is identified as OOD if its distances to the class prototypes are much greater than the support radii of all known classes. The support radius of a class is defined as the maximum distance from intra-class samples to the prototype. The probability-based approach identifies a sample as OOD if its probabilities of being any support classes are small.

*4.4.1 Distance-based Discriminator.* The prototype $\bar{\mathbf{x}}^c$ for class $c \in C$ in the support set is calculated as follows [35]:

$$\bar{\mathbf{x}}^c = \frac{1}{K} \sum_{i=1}^K f_\Theta(s_i^c), \quad (6)$$

where $s_i^c \in \mathcal{S}_c$ is a sample with a ground-truth label in the support set. Since intra-class samples are supposed to be clustered in the latent space, a normal sample in the query set will be close to the prototype of its proper category, while an OOD sample will be far away from all clusters. We define the radius of class $c$ as the maximum distance from any samples in class $c$ to $\bar{\mathbf{x}}^c$:

$$\gamma_c := \max_i \{dist\left(\bar{\mathbf{x}}^c, f_\Theta(s_i^c)\right)\}, i = 1, \ldots, K, \quad (7)$$

where $dist(\bar{\mathbf{x}}^c, f_\Theta(s_i^c))$ is the distance between $\bar{\mathbf{x}}^c$ and $f_\Theta(s_i^c)$. A query sample is supposed to be an OOD sample if its distance to $\bar{\mathbf{x}}^c$

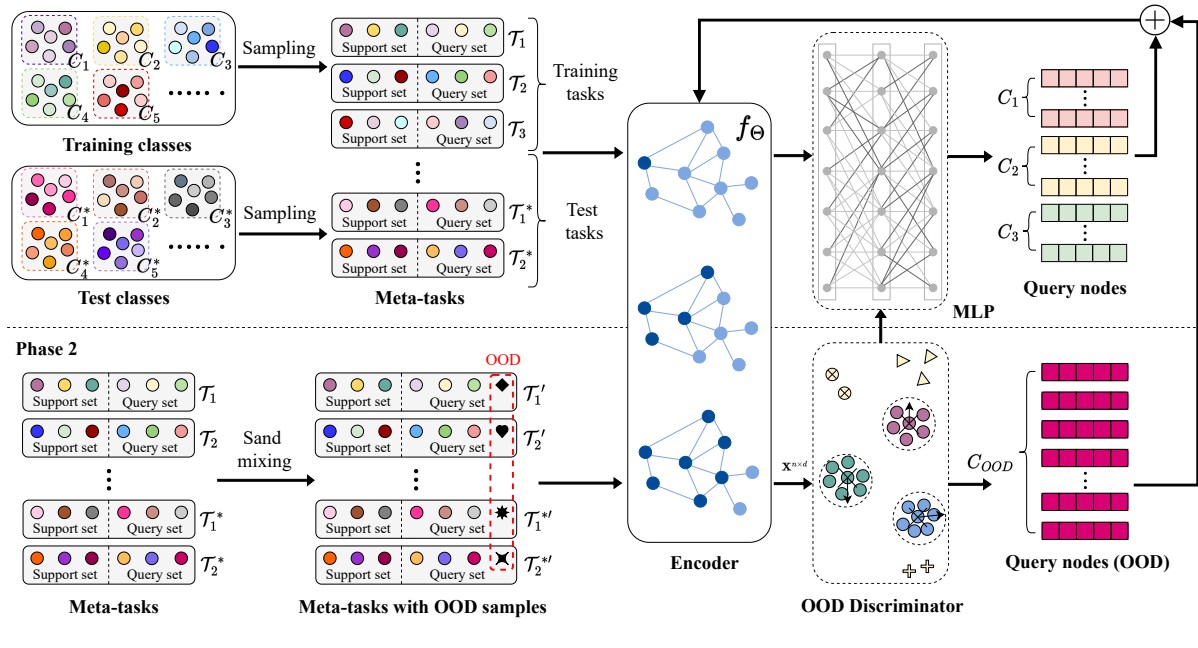

**Figure 1: An overview of the SMUG framework.**

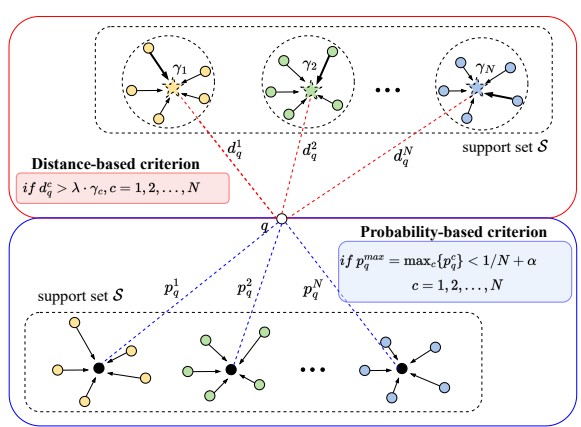

**Figure 2: Illustrations of the OOD detection criteria.**

is much larger than $\gamma_c$. We calculate the distances between $f_\Theta(q)$ and the prototypes of every class in a support set:

$$\mathbf{dist}_q := \left[ dist_q^1, dist_q^2, \ldots, dist_q^N \right], \qquad (8)$$

where $dist_q^c = dist(\bar{\mathbf{x}}^c, f_\Theta(q))$ is the distance from $f_\Theta(q)$ to $\bar{\mathbf{x}}^c$. Formally, we identify the test sample $q$ as an OOD sample if

$$dist_q^c > \lambda \cdot \gamma_c, \qquad (9)$$

for $\forall c \in \mathcal{C}$, where $\lambda$ is an adjustable parameter.

Conventional GFSC models are designed to maximize the distances between inter-class samples and minimize the distances of intra-class samples. In this work, we develop a novel loss function to enable the encoder to distinguish OOD samples from normal ones. The motivation of our method is to feed the results of the OOD discriminator back to the encoder. Specifically, we set up a reward $r$

to assess the performance of the discriminator, which represents the accuracy of identifying OOD samples in the current meta-task $\mathcal{T}$. Moreover, the encoder is expected to push OOD samples far away from normal ones. Therefore, the loss of the OOD discriminator with the distance-based criterion is defined as follows:

$$L_{OOD-dist} = -r \times \overline{dist}, \qquad (10)$$

where $r = TP/\Omega$ is the detection accuracy of the OOD discriminator, $TP$ is the number of OOD samples in $Q^{OOD}$ being correctly identified by the discriminator, $\Omega$ is the number of artificial OOD samples, $\overline{dist} = \frac{1}{N \times \Omega} \sum_{c=1}^N \sum_{i=1}^\Omega (dist_i^c - \gamma_c)$ is the average difference between $dist_i^c$ and $\gamma_c$ of all classes, and $dist_i^c$ is the distance between $\bar{\mathbf{x}}^c$ and $f_\Theta(q_i^{OOD})$. Minimizing $L_{OOD-dist}$ forces OOD samples to be far away from normal samples and increases the performance of the OOD discriminator.

*4.4.2 Probability-based Criterion.* The distance-based discriminator identifies an OOD sample by considering the absolute distances of a sample to the prototypes of every class in the support set. It ignores the relative positions of the clusters in the support set. We propose a probability-based scheme to address this difficulty. The primary motivation is that the probability of an OOD sample being classified into any normal class will be small and even. In this scheme, we use the distance between the query sample $q$ and the prototype $\bar{\mathbf{x}}^c$ to deduce the probability of being divided into a class $c$. Specifically, we calculate the distances between $f_\Theta(q)$ and the prototypes $\bar{\mathbf{x}}^c$ for each class $c$ according to Equ. (8). Then, the probabilities of $q$ being classified into these classes are calculated as follows:

$$\mathbf{p}_q = softmax(\mathbf{dist}_q) = [p_q^1, p_q^2, \ldots, p_q^N], \qquad (11)$$

where $p_q^i = \frac{\exp(-dist_q^i)}{\sum_{j=1}^N \exp(-dist_q^j)}$. Then, the maximum probability of $q$ being classified into a normal class is calculated as follows:

$$p_q^{max} = \max_i \{p_q^i\}. \qquad (12)$$

A query sample $q$ is supposed to be an OOD sample if

$$p_q^{max} < 1/N + \alpha, \qquad (13)$$

where $N$ is the number of classes in the query set, and $\alpha$ is an adjustable parameter. An adaptive threshold is adopted in this criterion. Accordingly, the probability-based loss of the OOD discriminator is formulated by:

$$L_{OOD-prob} = -\frac{1}{\Omega} \sum_{q \in Q^{OOD}} \exp\left(1 - p_q^{max}\right). \qquad (14)$$

where $\Omega$ is the number of OOD samples in a meta-training task. Minimizing the loss forces the maximum probability of dividing an OOD sample into any known classes to be small.

*4.4.3 Model Training.* We first train the encoder $f_\Theta$ and the classifier $g_{\Theta'}$ with meta-tasks without sand mixing, i.e., only normal samples are used for the GFSC model training. We use the cross-entropy loss as $L_{FSC}$ for model training in this phase. This phase consists of 1000 training episodes, and each episode performs a meta-training task. We perform 50 meta-validation and meta-test tasks every ten episodes.

The second phase fine-tunes the trained encoder $f_\Theta$ based on meta-tasks with mixed OOD samples. This phase comprises 500 episodes, and each episode performs one meta-training task. We execute 100 meta-test tasks every 50 episodes. Moreover, this phase only trains the encoder $f_\Theta$ and freezes the parameters of the classifier $g_{\Theta'}$. The encoder is trained with a combination loss as follows:

$$L = L_{FSC} + \beta L_{OOD-*}, \qquad (15)$$

where $\beta$ is an adjustable hyper-parameter to trade-off between the two kinds of losses, $L_{FSC}$ is the FSC loss function as presented in Equ. (1), and $L_{OOD-*}$ represents the loss based on the feedback from the OOD discriminator, $* \in \{dist, prob\}$ indicates different discriminators. Additionally, this work employs the Euclidean distance to calculate the distances of samples in the latent space. We use the Adam gradient descent method to update the model parameters.

## 5 EXPERIMENTS

This section presents the experiments conducted to evaluate the effectiveness of our method. All experiments are implemented based on the Pytorch framework. The source code of this paper is available at https://anonymous.4open.science/r/SMUG-89E1.

## 5.1 Experimental Settings

*5.1.1 Datasets.* We adopt four graph datasets to evaluate the performance of the proposed method. Table 1 presents a summary of the datasets. The detailed descriptions of the datasets are as follows:

**Amazon-Clothing (Ama-C) [52]** is a product graph built with the products in "Clothing, Shoes, and Jewelry" on Amazon. Each product is considered a node, and its description is used to construct the node attributes. The substitutable relationship ("also viewed") creates links between products.

**Amazon-Electronics (Ama-E) [52]** is another Amazon product graph containing products in "Electronics". Similar to Ama-C, each node denotes a product, and its attributes are retrieved from the product description. The complementary relationship ("bought together") between products creates the edges.

**Cora [53]** is a citation graph where nodes are scientific papers. Each paper cites or is cited by at least one paper. That is, there is at least one connection for a node. Each node is associated with a one-hot word vector to indicate its topics.

**DBLP [54]** is also a citation graph where each node represents a paper, and the edges represent their citation relations. Paper abstracts are used to construct node attributes. The published venue defines the label of a node.

**Table 1: A summary of the datasets.**

| Datasets | # samples | # edges | # attrs | # labels |
|----------|-----------|---------|---------|----------|
| Ama-C    | 24,919    | 91,680  | 9,034   | 77       |
| Ama-E    | 42,318    | 43,556  | 8,669   | 167      |
| Cora     | 2,708     | 5,429   | 1,433   | 7        |
| DBLP     | 40,672    | 288,270 | 7,202   | 137      |

We randomly select a proportion $\eta$ of classes as unobserved since the above datasets do not contain any ground-truth OOD samples. Then, we use 50% of the selected samples as artificial OOD samples for sand mixing. The remaining 50% of the selected samples are used as test samples to evaluate the accuracy of the OOD discriminators. These selected OOD samples are isolated from the original datasets, and the remaining classes are used to evaluate the classifier's performance. We evaluate the performance of node classification tasks based on two few-shot settings: 5-way 5-shot (5, 5) and 10-way 5-shot (10, 5). Due to the category limitation of Cora (it has only seven categories), we adjust the settings to 3-way 5-shot (3, 5) and 3-way 3-shot (3, 3). The query size is the same as the support size in all experiments. We adopt two widely used metrics, including Accuracy (ACC) and Micro-F1 (F1), to evaluate the performance of the classification task. The conventional metric Area Under the Receiver Operating Characteristic curve (AUROC) is used to evaluate the accuracy of OOD discriminators. We repeat each experiment 20 times and report the average of the results.

*5.1.2 Baseline Methods.* We compare SMUG with several state-of-the-art methods for few-shot learning in attribute graphs, including GCN [37], Meta-GNN [55], AMM-GNN [6] and GPN [3]. To evaluate the performance of OOD detection, we compare SMUG with EGM [11], GOOD [12], OEC [9], and GROOS [14]. The detailed descriptions of these methods are as follows:

- Methods for node classification of attribute graphs
  **GCN [37]** learns node representations based on the recursive aggregation of neighbors' features.
  **Meta-GNN [55]** obtains the prior knowledge of classifiers by training on many similar few-shot learning tasks to tackle the few-shot node classification problem in graph meta-learning settings.
  **AMM-GNN [6]** leverages an attribute-level attention mechanism to learn more effective transferable knowledge for meta-learning.
  **GPN [3]** derives a generalized model based on meta-learning for attribute node classification tasks.
- Methods for OOD detection
  **EGM [11]** detects OOD samples by identifying inconsistencies between activity patterns and predicted classes.

**GOOD [12]** uses the interval bound propagation (IBP) to derive a provable upper bound on the confidence of a classifier for OOD detection.

**OEC [9]** solves the OOD detection problem in few-shot learning by adapting conventional OOD methods for standard classification settings.

**GROOS [14]** uses a generic vector to represent OOD samples and identifies an OOD sample by its distance to the vector.

*5.1.3 Hyper-parameters Settings.* We set the proportion $\varphi$ of meta-training tasks in our experiments to 0.2. We set $\eta = 10\%$ to select artificial OOD samples for the best OOD sample detection. In the distance-based discriminator, we set the effects of the distance coefficient $\lambda = 1.2$. Similarly, we set $\alpha = 0.35$ in the probability-based discriminator, and $\beta = 0.25$ to balance $L_{FSC}$ and $L_{OOD-*}$. Default parameters are used for baseline methods.

## 5.2 Effectiveness Evaluation

This section evaluates the performance of SMUG in node classification and OOD detection tasks.

*5.2.1 Node Classification.* Table 2 shows the classification performance of different methods, with the best performance highlighted in bold. $\text{SMUG}_d$ uses the distance-based discriminator, while $\text{SMUG}_p$ employs the probability-based discriminator.

The results show that GFSL-based methods such as SMUG, AMM-GNN, and GPN have much higher classification performance than the other approaches, indicating the effectiveness of GFSL models. Moreover, $\text{SMUG}_d$ and $\text{SMUG}_p$ achieve higher performance than GPN, verifying that introducing OOD detection improves the performance of node classification tasks. In particular, SMUG achieves 82.3% ACC and 81.4% F1 on DBLP for the 5-way 5-shot setting, making an improvement of 2.2% and 1.6% over GPN, respectively. Besides, $\text{SMUG}_p$ obtains slightly higher accuracy than $\text{SMUG}_d$, indicating that the probability-based discriminator performs better than the distance-based one. An interpretation is that the probability-based discriminator employs an adaptive threshold to determine the OOD samples.

*5.2.2 Out-of-distribution Detection.* This section evaluates the performance of SMUG in detecting OOD samples versus $\eta$, which is the proportion of classes selected for artificial OOD samples. Fig. 3 shows the results based on DBLP. Similar results are obtained for other datasets and omitted for space concerns. SMUG performs best in all OOD detection tasks and achieves the highest AUROC (83.7%) when $\eta$=10%. AUROCs of the baseline methods decrease gradually as $\eta$ increases, while the performance of SMUG increases first and then decreases slightly. This is because SMUG can learn the distributions of OOD samples based on the feedback from the discriminators. Adding more OOD samples can benefit the training of the encoder. However, a large proportion of OOD samples will degrade the performance of the discriminators because fewer normal samples in a meta-task are used for model training, reducing the effectiveness of the encoder.

## 5.3 Parameters Sensitivity

We also conduct experiments to explore the effects of the distance coefficient $\lambda$ in the distance-based discriminator, the probability

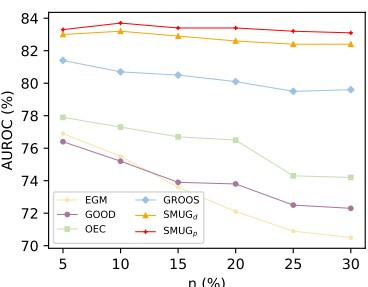

**Figure 3: Performance of OOD detection.**

coefficient $\alpha$ in the probability-based discriminator, the coefficient $\beta$ of SMUG, and the proportion $\varphi$ of meta-training tasks based on DBLP. Fig. 4(a) shows that AUROC first increases with the increase of $\lambda$ and peaks at $\lambda = 1.2$. This finding is reasonable because the radius of a class is approximately fixed for a given dataset. A smaller $\lambda$ will lead to many normal samples being mistakenly identified as OOD, while a larger $\lambda$ will lead to many OOD samples mistakenly classified as normal. This conclusion is also applicable to the behavior of $\alpha$ and $\beta$, as shown in Fig. 4(b) and (c). If the OOD loss has a larger weight with the increase of $\beta$, the encoder has a higher priority to distinguish OOD samples from normal ones. On the other hand, large values of $\beta$ will harm the classification of normal samples, leading to the peaking behaviors of AUROC and F1 for node classification. The similar effect acts on the proportion $\varphi$. Overall, ACC and F1 scores have similar behaviors versus the changes in the four parameters. The peaking behavior of both metrics verifies the mutual benefit between OOD detection and node classification. We obtain similar results for $\text{SMUG}_d$ on other datasets, which are omitted for space concerns.

## 5.4 Ablation Experiments

We investigate the effects of the OOD discriminator based on DBLP. We implement five variants of SMUG, including SMUG-SM, SMUG+EDC, SMUG+MDC, SMUG+CDC, and SMUG+PC. SMUG-SM is a version of SMUG without sand mixing. It is a normal graph few-shot learning model trained with clean meta-training tasks. SMUG+EDC is a version of SMUG with a distance-based discriminator that employs the Euclidean distance. SMUG+MDC adopts the Mahalanobis distance, and SMUG+CDC uses Cosine Similarity for distance assessment. SMUG+PC refers to the version of SMUG using the probability-based discriminator.

Fig. 5 demonstrates that SMUG-SM achieves the lowest OOD detection accuracy and node classification accuracy. An interpretation is that the graph encoder only learns prior knowledge about normal samples if sand mixing is not performed. This finding verifies the effectiveness of sand mixing because it enables the model to learn prior distributions of OOD samples. SMUG+EDC, SMUG+MDC, and SMUG+CDC achieve much higher performance than SMUG-SM, verifying the effectiveness of the distance-based discriminator in OOD detection. These three versions have comparable OOD detection and node classification performance, indicating limited impacts of distance functions on the performance of OOD detection. SMUG+PC obtains the best performance in both OOD detection

Table 2: Classification performance of different methods.

| Method | Ama-C | | | | Ama-E | | | | Cora | | | | DBLP | | | |
|---|---|---|---|---|---|---|---|---|---|---|---|---|---|---|---|---|
| | (5, 5) | | (10, 5) | | (5, 5) | | (10, 5) | | (3, 5) | | (3, 3) | | (5, 5) | | (10, 5) | |
| | ACC | F1 | ACC | F1 | ACC | F1 | ACC | F1 | ACC | F1 | ACC | F1 | ACC | F1 | ACC | F1 |
| GCN | 59.3 | 56.6 | 44.8 | 40.3 | 59.6 | 55.3 | 47.4 | 48.3 | 75.1 | 74.2 | 65.8 | 64.4 | 68.3 | 66.0 | 51.2 | 47.6 |
| Meta-GNN | 77.3 | 77.5 | 64.2 | 62.9 | 67.9 | 66.8 | 60.8 | 60.1 | 76.7 | 75.2 | 67.5 | 66.1 | 78.2 | 78.2 | 68.1 | 67.2 |
| GPN | 78.6 | 79.0 | 67.7 | 68.9 | 70.9 | 70.6 | 62.4 | 63.7 | 77.1 | 76.3 | 69.7 | 69.4 | 80.1 | 79.8 | 69.0 | 69.4 |
| AMM-GNN | 79.5 | 77.3 | 69.6 | 67.2 | 71.4 | 68.8 | 63.3 | 61.0 | 73.7 | 71.0 | 69.9 | 66.9 | 79.1 | 77.6 | 55.7 | 45.6 |
| SMUG$_d$ | 78.7 | 78.3 | 69.5 | 68.9 | 69.9 | 69.5 | 63.4 | 63.8 | 78.2 | 77.4 | 69.8 | 68.2 | 79.9 | 79.3 | 69.1 | 68.6 |
| SMUG$_p$ | **79.9** | **79.1** | **70.6** | **70.3** | **72.3** | **72.0** | **64.4** | **64.1** | **79.2** | **78.9** | **70.5** | **70.1** | **82.3** | **81.4** | **71.3** | **71.0** |

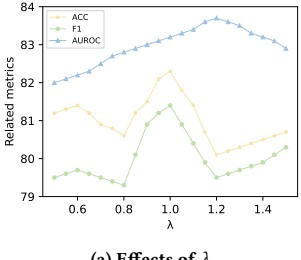

(a) Effects of $\lambda$

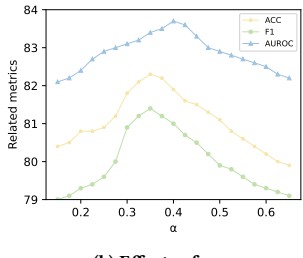

(b) Effects of $\alpha$

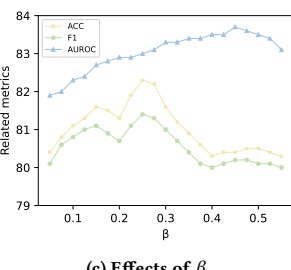

(c) Effects of $\beta$

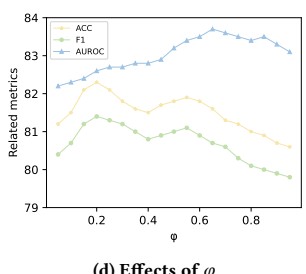

(d) Effects of $\varphi$

Figure 4: Effects of hyper-parameters.

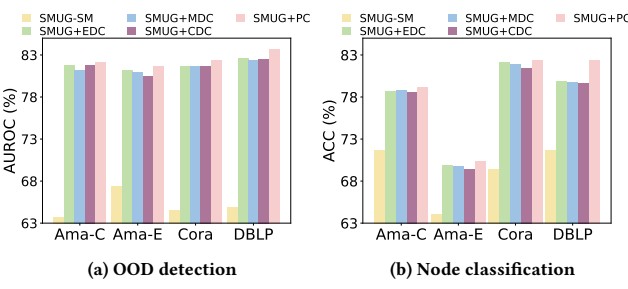

(a) OOD detection

(b) Node classification

Figure 5: Performance of different variants of SMUG.

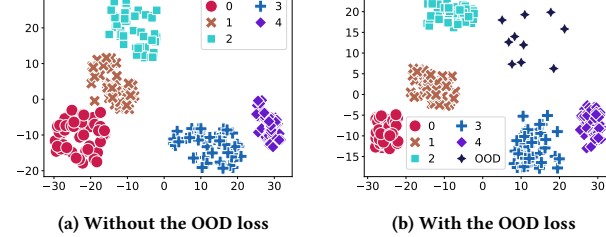

(a) Without the OOD loss

(b) With the OOD loss

Figure 6: Embeddings without and with the OOD loss.

and node classification. This is because the probability-based discriminator adopts an adaptive threshold for OOD identification, which is neglected by the distance-based criterion.

## 5.5 Case Study

To further display the promotion of OOD detection to GFSC, we randomly select two classes in Cora as OOD samples and treat the other five classes as normal. We plot the relative positions of node embeddings in the latent space with and without sand mixing. Node embeddings are mapped into a 2D space using t-SNE for visualization [56]. Samples in different categories are presented with different colors and shapes. Fig. 6a shows that samples of classes 0 and 1 are encoded closely without sand mixing. Moreover, all these clusters have relatively large support radii. In contrast, Fig. 6b shows that samples in normal classes are encoded more closely than those without sand mixing. Besides, the prototypes of different classes are encoded much further. Moreover, the encoder also successfully distinguishes OOD samples from normal ones.

## 6 CONCLUSION

This paper proposes a novel graph few-shot learning framework with sand mixing for OOD sample detection. It powers graph few-shot learning models with the ability to identify OOD samples and improves classification performance. To address the cold-start problem due to the lack of ground-truth OOD samples, we select several known classes as artificial OOD samples and mix them with normal samples in the query sets of meta-training tasks. We develop two discriminators to distinguish OOD samples from normal ones. Moreover, a combined loss is designed to optimize the encoder based on the performance of OOD discriminators. Experiments based on four benchmark datasets demonstrate that our method performs better than state-of-the-art methods in OOD detection and few-shot classification tasks. In future work, we would like to generalize the proposed few-shot learning framework to other fields, such as computer vision and natural language processing. Another interesting direction is to cluster detected OOD samples in dynamic graphs and study the evolution of community structures.

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

# APPENDIX A: THEORETICAL ANALYSIS OF THE PROPOSED METHOD

This section presents the theoretical analysis to verify that the proposed method can improve the classification performance of FSC models. We assume an $N$-way $K$-shot few-shot classification problem, which contains a small number of OOD samples. Given a sample-label pair $(\mathcal{X}, \mathcal{Y})$, our model attempts to learn a compressed representation of $\mathcal{X}$ by an encoder $f_\Theta : \mathcal{X} \to \mathbf{Z}^{n \times d}$. The encoder discards irrelevant information that does not contribute to the prediction of $\mathcal{Y}$, where $\mathcal{X} = \mathcal{X}^{ID} \cup \mathcal{X}^{OOD}$, $\mathcal{Y} = \mathcal{Y}^{ID} \cup \mathcal{Y}^{OOD}$.

We employ the Information Bottleneck principle [57] to analyze the model proposed for layer-wise analyses of Deep Neural Networks in terms of information compression efficiency [58]. Specifically, the mutual information $I(\cdot; \cdot)$ is employed to quantify the statistical dependence between two variables. For convenience, we use $X$, $Y$, and $Z$ to denote variables in the subspaces $\mathcal{X}$, $\mathcal{Y}$, and $\mathbf{Z}$. We

also abbreviate $I(X^{ID} \cup X^{OOD}; Z)$ as $I(X; Z)$. Accordingly, $I(X; Z)$ represents the amount of information maintained in $Z$, which is learned by the FSC encoder. Correspondingly, $I(Z; Y^{ID})$ represents the amount of relevant information about $Y^{ID}$ that is learned by the classifier. Then, the losses $L_{FSC}$ and $L_{OOD}$ can be formulated as:

$$L_{FSC} = I(X; Z) - I(Z; Y^{ID}), \tag{16}$$

and

$$L_{OOD} = I(X; Z) - I(Z; Y^{OOD}), \tag{17}$$

where variables $X$, $Y$, and $Z$ take values from the subspaces of $\mathcal{X}$, $\mathcal{Y}$, and $\mathbf{Z}$, respectively. Moreover, $I(X; Z) = I(X^{OOD}; Z) + I(X^{ID}; Z)$. Then, the proposed joint loss can be rewritten from the perspective of information theory as follows:

$$\begin{aligned} L &= (1+\beta)I(X; Z) - I(Z; Y^{ID}) - \beta I(Z; Y^{OOD}) \\ &= (1+\beta)I(X^{ID}; Z) - I(Z; Y^{ID}) \\ &\quad + (1+\beta)I(X^{OOD}; Z) - \beta I(Z; Y^{OOD}), \end{aligned} \tag{18}$$

where $\beta$ is an adjustable hyper-parameter. Further, $L$ can be divided into two parts:

$$L'_{FSC} = (1+\beta)I(X^{ID}; Z) - I(Z; Y^{ID}), \tag{19}$$

and

$$L'_{OOD} = (1+\beta)I(X^{OOD}; Z) - \beta I(Z; Y^{OOD}). \tag{20}$$

Since we are concerned with the classification effectiveness of FSC, we analyze $L'_{FSC}$ for further analysis. The mutual information can be rewritten as the Kullback-Leibler divergence between the marginal and conditional probability distributions as follows:

$$I(X^{ID}; Z) = \mathbb{E}_{x,z}[D_{KL}(p(z \mid x) \parallel p(z))], \tag{21}$$

where $\mathbb{E}(\cdot)$ is the expectation operation, and $D_{KL}(P \parallel Q) = \sum_{x \in X} P(x) \log \frac{P(x)}{Q(x)}$. The empirical distribution of $p(z)$ can be modeled as [58]:

$$p(z) = \frac{1}{N} \sum_{i=1}^{N} \delta(z - f_\Theta(x_i)), \tag{22}$$

where $\delta(\cdot)$ is the Dirac delta function, $x_i$ is the sample of the $i$-th class. The conditional distribution $p(z \mid x_i)$ can be formulated as an isotropic normal distribution around the observation in the embedded space:

$$\begin{aligned} p(z \mid x_i) &= \mathcal{N}(z \mid f_\Theta(x_i, \delta^2 I)) \\ &= \frac{1}{(2\pi\sigma)^{d/2}} \exp \frac{\parallel z - f_\Theta(x_i) \parallel^2}{2\sigma^2}, \end{aligned} \tag{23}$$

where $\sigma$ is the deviation caused by randomness. After substituting Equ. (22) and Equ. (23) into Equ. (21), we have:

$$\begin{aligned} &I(X^{ID}; Z) \\ =& \mathbb{E}_{x,z}\left[\log \frac{p(z \mid x)}{p(z)}\right] \\ =& \int \frac{1}{N} \sum_{i=1}^{N} \log \left[\frac{1}{(2\pi\sigma)^{d/2}} \exp - \frac{\parallel z - f_\Theta(x_i) \parallel^2}{2\sigma^2}\right] + \log \frac{1}{N} dx \\ =& \frac{1}{N^2} \sum_{z \in Z} \sum_{i=1}^{N} \left(-\frac{\parallel z - f_\Theta(x_i) \parallel^2}{2\sigma^2} + \log \sigma^d\right) + const, \end{aligned} \tag{24}$$

where $\parallel z - f_\Theta(x_i) \parallel^2$ is the distance from $z$ to $f_\Theta(x_i)$. Equ. (24) implies that the mutual information is relevant to the sum of distances between all pairs of samples in the embedding space. Meanwhile,

the class conditional probability over $z$ could be modeled as an isotropic normal distribution around the class centroid:

$$\begin{aligned} p(z \mid y) &= \mathcal{N}(z; \mu_y, \sigma_y I) \\ &= \frac{1}{(2\pi\sigma_y^2)^{d/2}} \exp \left(-\frac{\parallel z - \mu_y \parallel^2}{2\sigma_y^2}\right), \end{aligned} \tag{25}$$

where $\mu_y$ represents the class centroid and $\sigma_y$ denotes the deviation. $I(Z; Y^{ID})$ is then rewritten as follows:

$$\begin{aligned} &I(Z; Y^{ID}) \\ =& \mathbb{E}_{y,z}\left[\log \frac{p(z \mid y)}{p(z)}\right] \\ =& \int \frac{1}{N} \sum_{i=1}^{N} \delta(z - z_i) \log p(z \mid y) dy \\ =& \frac{1}{N} \sum_{y=1}^{N} K \log \frac{\exp\left(-\frac{\|z-\mu_y\|^2}{2\sigma_y^2}\right) - \log \sigma_y^d}{\sum \exp\left(-\frac{\|z-\mu_y\|^2}{2\sigma_y^2}\right) - \log \sigma_y^d} + const, \end{aligned} \tag{26}$$

where $\parallel z - \mu_y \parallel^2$ is the distance from $z$ to $\mu_y$. It is negatively correlated with class biases. To increase $I(Z; Y^{ID})$, the intra-class deviation, i.e., the distances between intra-class sample pairs in the embedded space, should be reduced. Meanwhile, the reduction of $I(X^{ID}; Z)$ is achieved by increasing the distances between inter-class sample pairs. Therefore, minimizing $L'_{FSC}$ forces pulling intra-class samples closer while pushing inter-class samples away. Compared with the original loss (16), $L'_{FSC}$ has $\beta$ times gains from OOD detection, which makes the distances of samples from different categories further. Thus, our method can enhance the classification performance of FSC models. Analogously, $L'_{OOD}$ could separate OOD samples from normal ones.

