# OpenReview forum: "SMUG: Sand Mixing for Unobserved Class Detection in Graph Few-Shot Learning"
_ACM.org/TheWebConf/2024/Conference — TheWebConf24 Oral_

### Official Review · Reviewer_YnZZ · 2023-10-31

**Novelty:** 5
**Technical Quality:** 5

**Review:**

This paper proposes the idea of combining graph few-shot classification with OOD detection. The authors' main contribution is to use the sand mixing trick to add OOD samples to the training phase, then leverage correspondingly crafted loss functions to work in a multi-task manner. In general, I think this paper works on an important problem, the task is well-motivated and clearly defined. The experiments are somewhat limited by the scope of datasets and the way OOD samples are synchronized, which is understandable for an academic lab. Indeed, stronger evidence needs to be examined in real-world applications. More detailed comments are referred to below.

**Questions:**

General comments
- This paper leverages the simple sand mixing trick to outperform baselines, code is provided for the sake of reproducibility.
- The paper is generally well written and easy to follow, the authors managed to dissect the methodology in a reader-friendly manner.
- I should point out that the distance-based criterion the authors propose, is somewhat connected to LDAM-loss [1], the key idea is to maximize the margin between well-classified samples and "OOD" samples, perhaps this work can be further improved in that regard.
- This work is inherently limited by the way its OOD samples are synchronized, I would prefer the authors can put a separate subsection to discuss this. Therefore, I believe we cannot safely assume such a technique can be directly applied, however, the community can still draw some helpful insights.

[1] Cao, Kaidi, et al. "Learning imbalanced datasets with label-distribution-aware margin loss." Advances in neural information processing systems 32 (2019).

**Reviewer Confidence:**

4: The reviewer is certain that the evaluation is correct and very familiar with the relevant literature

**Scope:**

4: The work is relevant to the Web and to the track, and is of broad interest to the community

---

### Official Review · Reviewer_7zmr · 2023-11-21

**Novelty:** 5
**Technical Quality:** 5

**Review:**

# Summary

This paper addresses the issue of out-of-distribution (OOD) samples in graph few-shot learning. It proposes to train an OOD discriminator using artificial OOD samples along with the node classification model. The experiments show superior classification and OOD detection power.

# Pros

* The authors propose two objectives for training OOD discriminators and perform thorough experiments on them.
* The idea of training OOD discriminators using artificial samples is nice and straightforward. It has great potential for tasks beyond graph learning.

# Cons

* While the authors mention future work, a more thorough discussion on the current limitations of SMUG would be beneficial.

**Questions:**

N/A

**Reviewer Confidence:**

4: The reviewer is certain that the evaluation is correct and very familiar with the relevant literature

**Scope:**

4: The work is relevant to the Web and to the track, and is of broad interest to the community

---

### Official Review · Reviewer_mnv1 · 2023-11-24

**Novelty:** 4
**Technical Quality:** 4

**Review:**

This paper proposes a graph few-shot classification framework for unobserved class detection. The proposed model presents a sand mixing scheme that introduces observed classes as artificial OOD samples into meta-tasks. Two OOD detection discriminators with corresponding loss functions are developed to optimize the graph-based encoder. Experiments are conducted on four datasets to show the effectiveness of the proposed method.
Strengths:
S1. The paper is easy to follow.
S2. The idea of incorporating the unsupervised OOD discriminator to detect OOD samples is interesting.
S3. The model is reproducible as the source code is provided.
Weaknesses:
W1: The logic in the introduction section is unclear and somewhat chaotic.
W2: The experiment is not convincing as the selected baseline methods are somewhat old.
W4: The experiment setting is not clear, especially the parameter setting of baselines.

**Questions:**

1.	Figures 1 and 2 are not readable, it is suggested to add more explanations in the captions.
2.	Why use default parameters for baseline methods instead of optimal ones? The baseline methods should be tuned to obtain the optimal performance.
3.	The compared methods are old, and more recent models need to be considered.
4.	Figure 3 can accommodate results from at least two datasets, it would be better to show more performance of the OOD detection task.
5.	Why all the OOD baseline methods are published before 2021? More recent approaches need to be considered.
6.	Table 2 shows that SMUGd does not have much advantage over the baseline GPN. Does this indicate that the OOD discriminator may not necessarily be effective?

**Reviewer Confidence:**

3: The reviewer is confident but not certain that the evaluation is correct

**Scope:**

3: The work is somewhat relevant to the Web and to the track, and is of narrow interest to a sub-community

---

### Official Review · Reviewer_x7Vk · 2023-11-26

**Novelty:** 5
**Technical Quality:** 5

**Review:**

This paper proposes to address the challenge of out-of-distribution sample detection in inductive graph few-shot classification (GFSC). The presented SMUG framework incorporates a sand mixing scheme to introduce observed classes as artificial OOD samples into meta-tasks, allowing the assessment of OOD discriminators. Two unsupervised OOD discriminators are designed to identify OOD samples based on distance and probability criteria. SMUG then optimizes the encoder by integrating the performance of OOD discriminators as a weak signal with the GFSC model. Experimental results on four datasets demonstrate the superior performance of SMUG in OOD detection and node classification over various baselines.


**Strengths:**

(+) The proposed setting that graph few-shot classification models may encounter unobserved classes during testing is practical and has wide applications.


(+) The idea of introducing observed classes as artificial OOD samples into meta-tasks is intuitive and reasonable. The proposed two unsupervised OOD discriminators are novel in my view.

(+) The evaluation protocol is comprehensive. Various baselines, including both GNNs and OOD approaches, are compared. The authors also conduct ablation studies and hyperparameter analyses to validate the contribution of their presented techniques and model configuration.

**Weaknesses:**

(-) The proposed framework is somehow complicated with many hyperparameters. It is unclear how the value of each hyperparameter is selected.

(-) Statistical significance tests are missing. It is unclear whether the gaps between SMUG and baselines/ablation versions are statistically significant or not. In fact, in Table 2 and Figure 5, some gaps are quite subtle, therefore p-values should be reported.

**Questions:**

- Could you explain how the value of each hyperparameter in Section 5.1.3 is selected? Do you have a validation set for this process?

- Why do you choose different types of edges to construct the graph for different datasets? To be specific, you use the substitutable relationship for Ama-C and the complementary relationship for Ama-E. Also, is there a way to extend SMUG to deal with multi-view networks?

- Could you conduct statistical significance tests to compare SMUG with the strongest baseline in Table 2 and each ablation version in Figure 5?

**Reviewer Confidence:**

2: The reviewer is willing to defend the evaluation, but it is likely that the reviewer did not understand parts of the paper

**Scope:**

3: The work is somewhat relevant to the Web and to the track, and is of narrow interest to a sub-community

---

### Decision · Program_Chairs · 2024-01-22

**Decision:**

Accept (Oral)

**Comment:**

The paper addresses the issue of out-of-distribution (OOD) detection. It uses a sand-mixing scheme, utilizing samples from observed classes as artificial OOD samples, (along with two unsupervised OOD discriminators developed in the paper) to improve graph few-shot learning tasks.

 The idea is simple but novel. As several reviewers noted, it may be more generally useful, as well as lead to additional insights. Although there were concerns about the thoroughness of the experiments, these issues were addressed in the rebuttal. Experiments show consistent results over SOTA.

 Strengths:
 * OOD detection is an important problem, and the technique is widely applicable.
 * The main idea is intuitive, with nice high-level motivation given.
 * Generally well written.

 Weaknesses:
 * There are many hyperparameters. This is handled using a validation set, but still adds additional complexity to the method.